# Multi-Sample Preparation Assay for Isolation of Nucleic Acids Using Bio-Silica with Syringe Filters

**DOI:** 10.3390/mi11090823

**Published:** 2020-08-30

**Authors:** Geun Su Noh, Huifang Liu, Myoung Gyu Kim, Zhen Qiao, Yoon Ok Jang, Yong Shin

**Affiliations:** Department of Convergence Medicine, Asan Medical Institute of Convergence Science and Technology (AMIST), University of Ulsan College of Medicine, 88 Olympicro-43gil, Songpa-gu, Seoul 05505, Korea; ngs90@hanmail.net (G.S.N.); liuhuifang.1229@gmail.com (H.L.); wws94@naver.com (M.G.K.); qiaozhen90@hotmail.com (Z.Q.); jangyo17@daum.net (Y.O.J.)

**Keywords:** multi-sample preparation assay, pathogen extraction, nucleic acid isolation

## Abstract

The spin-column system for the isolation of nucleic acids (NAs) from multiple samples presents the inconvenience of repeated experimentation, time-consumption, and the risk of contamination in the process of the spin-column exchange. Herein, we propose a convenient and universal assay that can be used to diagnose multiple pathogens using a multi-sample preparation assay. The multi-sample preparation assay combines a 96-well filter/membrane plate, a bio-micromaterial lattice-like micro amine-functional diatomaceous earth (D-APDMS), and homobifunctional imidoesters (HI) for the processing of pathogen enrichment and extraction for multiple samples simultaneously. The purity and quantity of the extracted NAs from pathogens (*E. coli* and *Brucella*) using the proposed assay is superior to that of the commercialized spin-column kit. The assay also does not require the replacement of several collection tubes during the reaction processing. For the multi-sample testing, we used as many as six samples simultaneously with the proposed assay. This assay can simultaneously separate up to 96 NAs from one plate, and the use of multichannel pipettes allows faster and simpler experimentation. Therefore, we believe it is a convenient and easy process, and can be easily integrated with other detection methods for clinical diagnostics.

## 1. Introduction

The entire world pays tribute to all medical staff working on the front lines of infectious disease diagnosis, especially during rapid virus and pathogen infections [1,2]. Currently, human infections caused by pathogens are spreading worldwide, and in particular, new and mutated pathogen infectious diseases can cause enormous social and economic losses [3,4]. Thus, rapid diagnostic techniques, combined with automation and/or a self-examination system, are increasingly expected to save labor as well as time and provide better protection of the medical inspectors [5].

In general, the spin-column based sample preparation technology is popular for nucleic acid (NA) extraction. However, it is an expensive kit and experienced technicians are required during the experiment. In addition, there is a possibility that NA contamination may occur in a complicated process [6]. Even if an experienced technician performs an experiment, NA extraction must be performed using one spin-column at a time; therefore, it is time-consuming to perform repeated experiments when many samples are present. If the time is shortened, the probability of NA contamination increases even if a skilled technician performs the extraction of many samples simultaneously. For NA extraction from multi-samples, the spin-column based sample preparation is unsuitable and labor intensive [7]. To address this issue, we used 96-well filter/membrane plates to extract NAs from multiple samples simultaneously. The 96-well filter/membrane plates have a long history as a reliable tool with parallel sample testing for the advancement of life science research and drug discovery. Since the micro 96-well plate was first developed for the flu pandemic in the 1950s by Takatsy [8], the development of the 96-well plate has never been introduced or even extended to multiple fields, typically working in cell culture, immune fluorescence diagnosis, NA amplification [9]. A comprehensive system, which could integrate innovative automated instruments such as a microplate reader, real-time quantitative PCR (qPCR), and autosampler is expected in terms of human–animal–ecosystems interfaces. Using a combination of a polyvinylidene fluoride (PVDF) membrane [10,11] plate and 96-well cell culture plate (F-Type), impurities that pass through the PVDF membrane plate come out alone on the 96-well cell culture plate; thus, they can be simply removed by pipetting. Commercialized plates are sterilized and packaged individually; therefore, they can be employed immediately in experiments without any need for preprocessing and can be easily used by anyone. Additionally, multichannel pipettes can be more convenient for conducting experiments on various samples simultaneously.

Meanwhile, our laboratory has paid increased attention to diatomaceous earth (DE) as reported in a previous study [12]. Diatomaceous earth [13,14,15] refers to soil formed by deposits of fine gray silicate components composed of the remnants of diatoms, which are primarily composed of silicic acid (Si(OH)4). They are known to be highly absorbent and resistant to heat at high temperatures with white and grayish colors and small pores. By utilizing these various advantages, DE was used in numerous areas such as filtration, sorbent, and drug delivery [16,17]. High performance of DE is expected in many areas of research because of its large surface area, volume ratio, and greater absorption when combined with other materials. Diatomaceous earth [18], which is used as a major material for the enrichment and NA extraction of pathogens and is easy to store because it can be stored for long periods at room temperature (RT).

In this study, we developed a convenient multi-sample preparation assay by uniting DE having micro-grid structure and 96-well filter/membrane plates, which could enrich the target samples for accurate diagnosis. A PVDF membrane plate and a 96-well cell culture plate (F-Type) are used to build an up–down combination spatial structure, which allows easier sample collection. The modified DE with micro-grid structure has been used as an enrichment medium for capturing the samples. The purpose of this study was to develop a technology that can detect pathogens with high sensitivity and process multiple samples simultaneously. Well-washed DE with micro-grid structure was treated with amine [19] by 3-aminopropylidimethylsilane (APDMS) [20,21,22] to use the DE for diagnostic purposes; the active DE surface supplies a large surface area for further sample enrichment. The positive charge of the amine-modified DE surface can bind with the negative charge of pathogen surface by electrostatic interaction [23]. Meanwhile, an additional homobifunctional imidoester (HI) reagent was used to combine more powerfully with NAs. The amine group of the HI surface can bind with the amine group of the NA by covalent bonding [23,24]. Globally, animal pathogens such as *Brucella ovis* and *Escherichia coli* [25] are causing huge financial losses and high incidence rates. These two pathogens were used to assess the performance of our developed methods. The combination system (AD–DMP filter plate system) comprising amine-functional diatomaceous earth (D-APDMS), dimethyl pimelimidate dihydrochloride (DMP), which is a type of HI, and 96-well filter/membrane plate can isolate at least six samples simultaneously, and has the ability to simultaneously isolate additional samples. Unlike commercialized NAs kits consisting of column and collet tube, the proposed assay requires minimal time because multiple samples can be simultaneously extracted. Further, it is economical, because there is no need to exchange the tube during the experiment. The purchase of additional experimental instruments is not required, because only pipettes are needed during the experiment, thus making it easily accessible for various research facilities. The proposed assay shows at least 10 times greater sensitivity as a result of real-time qPCR compared to commercialized NA kits and can achieve satisfactory results with laboratory equipment used by commercialized NA kits.

## 2. Experimental Section

### 2.1. Chemicals and Reagents

The MultiScreen_HTS_ HV 96-well filter plate (hydrophilic PVDF membrane with 0.45 µm pore size; Millipore Corporation, Burlington, MA, USA) and the MultiScreen Solvinert 96-well filter plate (hydrophobic Polytetrafluoroethylene (PTFE) with 0.45 µm pore size; Millipore Corporation) were purchased from Merck KGaA (Darmstadt, Germany). The 96-well cell culture plate (F-Type) was purchased from SPL Life Sciences (Gyeonggi-do, Korea). DE (flux-calcined powder), APDMS (97%), DMP powder, lysozyme from chicken egg white (lyophilized powder, 50 mg/mL in distilled water) were purchased from Sigma-Aldrich (St. Louis, MO, USA). Toluene (C_6_H_5_CH_3,_ 99.5%) and iso-propyl alcohol ((CH_3_)_2_CHOH) were purchased from Duksan-science (Seoul, Korea). Milli-Q water, ethanol (99%), and phosphate-buffered saline (PBS, 10×, pH 7.4) were ordered from Thermo Fisher Scientific (Waltham, MA, USA). Proteinase K solution (>600 mAU/mL), generally used in biology to digest protein, was purchased from Qiagen (Hilden, Germany). Defibrinated sheep’s blood was purchased from KisanBio (Seoul, Korea). All the reagents were used without any further purification and processing.

### 2.2. Instruments

D-APDMS nanomaterials were examined using a scanning electron microscope. (SEM; JEOL JSM-7500F, JEOL Ltd., Tokyo, Japan). The commercial QIAamp DNA Mini Kit (spin column) was used for NA isolation (DNA Mini Kit, Qiagen, Hilden, Germany). The spin-down device centrifuge (CF-5, 100~240 Vas, 50/60 Hz, 8 W), vortex mixer (T5AL, 60 Hz, 30 W, 250 V), and LABOGENE 1730R (220 V~, 60 Hz, 2.0 kVA, LABOGENE, South Korea), MSH-30d stirring heaters were produced by Daihan Scientific Co., Ltd. (Wonju-Si, Korea). The 15 mL H 20015 and 50 mL H 20050 Conical Tubes (HYUNDAI Micro co., Ltd, Seoul, Korea); the centrifuge (5810R, Eppendorf, Hamburg, Germany); CFX96 Touch Real-Time PCR detection system (BIO-RAD, Hercules, CA, USA); GeneAmp PCR system 9700 (LSK Singapore); electrophoresis apparatus (Submerge-Mini WSE-1710, ATTO, Daejeon, Korea); Dyne LoadingSTAR (DYNEBIO, Seongnam, Korea); Gel-Documentation System (Clinx Science Instruments, Shanghai, China) and NanoDrop 2000 (VWR International, Radnor, PA, USA) were used for the NA detection test.

### 2.3. Preparation of Amine-Functional DE

D-APDMS nanomaterials are pre-prepared to be used as a matrix in the enrichment and extraction process [26]. Referring to existing lab protocols [23,24], a new method was devised to apply to new systems. The DE was washed at RT for 10 min with distilled water (DW) as a dynamic stirring. After the stirring, it remained at RT for 1 min. Next, we carefully moved the supernatant—except sediment—to the new tube to obtain clean, even diatoms. The DE was cleaned again at RT with ethanol (EtOH, 99%) with dynamic stirring for 10 min. The washed diatoms are moved to a new tube and stored in a 65 °C dry oven for one d. Subsequently, 3 mL of APDMS was pipetted dropwise into 50 mL 95% (*v*/*v*) ethanol solution, and 1 g of washed DE was added with dynamic stirring. The reaction was maintained at RT for 4 h. The D-APDMS was washed with DW and then dried in a 65 °C dry oven overnight. Two previously reported protocols [20,21] were considered to compare with the newly designed protocol. The process of creating a washed diatom before the D-APDMS treatment is the same. It was then dynamically stirred in 25 mL of toluene and washed in 2 g of DE for 1 h at 65 °C. Next, it was placed in 500 μL of DW and stirred for 1 h; 1 mL of D-APDMS was then added and stirred. Subsequently, it was washed with iso-propyl alcohol and toluene and stored overnight in a dry oven at 65 °C; the dried D-APDMS was stored at RT until further analysis.

### 2.4. Biological Samples Cell Culture

*Brucella ovis* (ATCC 25840) and *Escherichia coli* (ATCC 25922) were used to assess pathogen diagnosis. After blending Brucella Broth Powder, Bacto^TM^ Agar powder, and DW, *B. ovis* was grown in a medium containing 5% defibrinated sheep’s blood [27]. The medium was incubated 48 to 72 h at 37 °C in an incubator that maintains CO_2_ atmosphere. *E. coli* was cultured at 37 °C in a medium mixed with Nutrient Broth Powder, Bacto^TM^ Agar Powder, and DW. After cultivating bacterial suspension, it was quantified in a medium mixed with Bacto^TM^ Agar powder and DW and diluted to a different concentration using PBS, 10×, pH 7.4.

### 2.5. Filter/Membrane-Based Pathogen Extraction NA

D-APDMS and DMP were used as the enrichment and extraction matrices for PBS with samples. First, the MultiScreen Solvinert 96-well filter plate and the 96-well cell culture plate (F-Type) are combined. Twenty microliters of D-APDMS suspension (50 mg/mL in DW) and 50 μL of DMP solution (100 mg/mL in DW) were pipetted into a sample solution. The NAs isolation was subsequently performed in the same PTFE membrane plate. Subsequently, 20 μL of Proteinase K, 150 μL of internal lysis buffer (100 mM Tris-HCl (pH 8.0), 10 mM ethylenediaminetetraacetic acid, 10% Triton X-100, and 1% sodium dodecyl sulfate), 30 μL of lysozyme solution (50 mg/mL in DW) were added separately. After mixing, the PVDF membrane plate was incubated in a dry oven for 30 min at 56 °C for DNA extraction. Operating the 500 relative centrifugal force (RCF) centrifuge for 2 min will allow it to exit the PTFE membrane plate with the exception of the NAs template fixed to the ADE through the DMP crosslinking. Using a multi-pipette, the remaining pellets in each hole were washed twice with a 300 μL PBS. For reverse crosslinking, 100 μL of elution buffer (10 mM sodium bicarbonate, pH > 10, adjusted by NaOH) was inserted into each hole. The PTFE membrane plate was kept at RT for 1 min and centrifuged for 2 min in the 500 RCF. The NAs template that eluted to the 96-well cell culture plate (F-Type) was moved to the tube using a pipette. After the supernatant containing the isolated DNA was stored at −20 °C until needed. The same sample (*B. ovis* in PBS, 10^5^ colony formation unit (CFU)) was considered as a positive control group using a commercial kit (QIAamp DNA Mini Kit, Qiagen) according to the manufacturer’s protocol. The maximum capacity of this commercialized kit was considered to be 200 μL. Evaluation and optimization of the NAs extraction experiments using DNA absorbance and real-time qPCR was performed.

### 2.6. Filter/Membrane-Based Pathogen Enrichment and Extraction NA

A 1 mL sample (*E. coli* in PBS) was prepared in the tube. As with the extraction procedure, 20 μL of D-APDMS suspension (50 mg/mL in DW) and 50 μL of DMP solution (100 mg/mL in DW) were pipetted into a sample solution. The 1 mL sample was shaken by hand for 5 min at RT. Since the maximum recommended capacity for the PVDF membrane plate manual is 300 μL, 250 μL of the mixture was transferred to the PVDF membrane plate. The 500 RCF centrifuges were used for 2 min. All 1 mL of the samples were moved repeatedly. In each hole containing the sample, 20 μL of Proteinase K, 150 μL of internal lysis buffer, 30 μL of lysozyme were added in order. After pipetting the samples and the mixture, the PVDF membrane plates were incubated for 30 min in a dry oven at 56 °C. After incubation of the sample, the PVDF membrane plate was operated for 2 min with a 500 RCF centrifuge, and the supernatant was removed from the 96-well cell culture plate (F-Type) excluding DNA extracted from pathogens. The 300 μL of PBS was placed in each hole on the PVDF membrane plate and the washing process was performed twice. The remaining pellets in each hole were elution buffered in 100 μL of elution buffer. After incubating it at RT for 1 min, it was operated in the 500 RCF centrifuge for 2 min. The DNA transferred to the 96-well cell culture plate (F-Type) was transferred to the tube. The NAs enrichment and extraction experiments were evaluated using qPCR.

### 2.7. Conventional and Real-Time PCR

The quality of enriched and extracted DNA was determined by conducting PCR and real-time PCR. The primers used are listed (Table 1). The heat circulation process used in PCR is as follows: An initial denaturation step at 95 °C for 15 min; 40 cycles at 95 °C for 30 s, 58 °C for 30 s, and 72 °C for 30 s and a final extension step at 72 °C for 7 min. Then, 5 µL of DNA was amplified in a total volume of 25 μL containing PCR buffer (10×, Qiagen), 2.5 mM MgCl2, 0.25 mM deoxynucleotide triphosphate, 25 pM of each primer, one unit of Taq DNA polymerase (Qiagen), and DW. The PCR products were separated by gel electrophoresis on 2% agarose gels containing Dyne LoadingSTAR and imaged with a Gel-Doc System (Clinx Science Instruments, Shanghai, China). The real-time PCR procedure was modified from the BIO-RAD real-time PCR Instrument protocol (BIO-RAD, Hercules, CA, USA). Briefly, 5 µL of isolated DNA was amplified in a total volume of 20 μL containing 2× Brilliant III SYBR Green QPCR master mix, 25 pM of each primer, and DI water. An initial pre-denaturation at 95 °C for 15 min was followed by 40 cycles at 95 °C for 10 s, 58 °C for 20 s, and 72 °C for 20 s, and by a final extension step at 95 °C for 10 s.

## 3. Results and Discussion 

### 3.1. Development of the Multiplex Enrichment, Extraction Method

As reported in previous studies, DE—with numerous qualities and various morphologies—is a good candidate for many bioengineering applications and show a powerful host–guest interaction [28,29] that supports NAs enrichment and extraction. Here, we studied the special lattice-like micro DE frame (Figure 1), the uniform lattice frame makes the DE in delicate micro 3-dimensional (3D) structure. The D-APDMS (Figure 1) has been developed and optimized to increase the DE surface activity and could support cell capturing, NAs enrichment, and extraction through powerful interaction and a large surface area. The schematic diagram of pathogen enrichment with the micro lattice-like D-APDMS and HI reagent, in the tube is shown in (Figure 1i). The imidoesters at both ends of the DMP help to form an amide bond with the amine group exposed on the surface of the D-APDMS and also support to combine pathogens or NAs. After the tube enrichment, the 96-well filter/membrane plate system has been used to do the multiplex NAs extraction study. The trapped pathogens are lysed in the 96-well filter/membrane plate (Figure 1ii). Subsequently, the exposed NAs combine with D-APDMS and DMP, and the NAs could then be chemically isolated through a reversible crosslinking reaction (Figure 1iii) [30]. In the case of extraction experiments, the mixture does not need to be moved; however, when enrichment and extraction are carried out at the same time, the mixture should be moved from the tube to the plate. However, after moving to the plate, multiple samples can be easily extracted at the same time using a multi-pipette. Additionally, there is no need to exchange tubes repeatedly, and up to 96 samples are available simultaneously.

### 3.2. Comparison of Amine-Functional Diatomaceous Earth (D-APDMS) Obtained via Two Methods and Optimization of Experimental Methods 

The DE amine-functionalized method in this study using toluene and iso-propyl alcohol is compared with the laboratory method using ethanol and DW. At the same time, the PBS washing volume was sequentially increased by 100 μL to improve purity more than the previously reported experimental method [23,24]. According to the results of the DNA absorbance in which NAs extraction was performed using two D-APDMS, the NA concentration was slightly decreased, but the average purity was the best when the PBS washing was 300 μL (Figure 2A). As a result of synthesizing the DNA absorbance results, it was confirmed that our D-APDMS method is more efficient among the two D-APDMS. It was also confirmed that our DE surface treatment method is simpler and more efficient, considering that another DE surface treatment method requires a number of reagents and temperature control of the D-APDMS process [20,21,22,23,24]. Experiments were conducted to optimize the amount of D-APDMS and DMP to fit filter/membrane plates. We compared the results through real-time qPCR. It was decided to apply D-APDMS 20 μL (50 mg/mL in DW) and DMP 50 μL (100 mg/mL in DW) according to the results of sequentially decreasing the amounts of D-APDMS and DMP used in the previously reported filter test method [12]. As a result, similar cycle threshold (C_T_) values were obtained when compared to previously reported NA separation methods and commercialized kits when the amounts of D-APDMS and DMP used in previously reported studies [12] were reduced to match the plate capacity (Figure 2B). When all the results were assembled, it was decided to select D-APDMS, which was surface-treated with ethanol and DW, and to reduce the D-APDMS and DMP to more than the amount reported in the previous study [12,24] when proceeding with the isolation of NAs. In addition, experiments were conducted to optimize several factors that influence NA isolation. In our system using D-APDMS (Figure 3A) and DMP, the pH value of the elution buffer is very important. Cross-linked DNA is efficiently released under high-pH conditions [30]. The C_T_ value results of qPCR of DNA extracted using buffers with different pH values were confirmed. Based on the confirmed results, pH values 10.5 and 10.6 do not show any special difference between them. Since the buffers (pH 10.5 and pH 10.6) were already very alkaline, it was confirmed that the crosslinking bond by the HI reagent was well separated and NAs extraction was also very effective. However, when checking the C_T_ value using qPCR we decided to use a buffer of pH 10.6 (C_T_: 24.147), showing a slightly lower C_T_ value than pH 10.5 (C_T_: 24.407) (Figure 3B). Another factor focused on was the presence or absence of lysozyme from the chicken egg white (lyophilized powder, 50 mg/mL in DW). As a result of checking using qPCR, it was confirmed that lysozyme does not affect the PCR process, because the C_T_ value with or without lysozyme is not significantly different (Figure 3C).

### 3.3. Comparison of Two Types of Filter/Membrane Plate

Based on the optimization of our convenient multiple NA extraction system, we have tested two types of the 96-well filter/membrane plate with different membranes (PTFE membrane and PVDF membrane plate). With the same filtration area (0.28 cm^2^) and same pore size (0.45 μm), both of them could carry the DE having micro-grid structure (2–10 μm) upon the membrane surface. First, the PTFE membrane plate is suitable for alcoholic solvents, strong acids, strong bases, gases, etc., excluding aqueous solutions. For example, distilled water and PBS, 10×, pH 7.4 cannot pass the PTFE filter. Notably, the extraction of the C_T_ values of the NAs proceeded by adding the D-APDMS and DMP had little difference from the C_T_ of the NAs extracted with a commercialized kit (QIAamp DNA Mini Kit (spin column)) (Figure 4A). In addition, we found that the pure PBS, 10×, pH 7.4 could not pass through the untreated PTFE filter, but when pathogen NAs extraction was performed, as treated with the lysis buffer which contained an organic solvent, the PTFE filter become wet and transparent for the PBS washing step. Secondly, the PVDF membrane plate is suitable for alcoholic solvents, strong acids, strong bases, gases, and aqueous solutions. Since the aqueous solution also passes through differently from the PTFE, PBS and distilled water can also pass through the plate. The same performance test was performed with the PVDF filter/membrane plate, the quantities of NAs are shown in Figure 4B. There was little difference between the C_T_ of NAs extracted with the D-APDMS and DMP, and the C_T_ of NAs extracted with commercialized kits. Above all, the distractions (sample and buffer) have been eliminated in both of the PTFE and PVDF plates; additionally, the DNA quantities from the “DNA of *E. coli* extracted using D-APDMS + DMP filter/membrane plate system.” Figure 4 revealed that DNA could be eluted well with both of those plates. However, due to the PTFE membrane being hydrophobic, as the PVDF membrane is hydrophilic, the active lattice-like micro D-APDMS fits the PVDF membrane better than the PTFE membrane. Therefore, the PVDF plate has been chosen to achieve higher flux for the convenient multiple NA extraction system.

### 3.4. Accuracy and Suitability of Multiple Diagnostic Systems that Concentrate and Extract Multiple Samples at a Time

To proceed with NAs enrichment and extraction, 1 mL of pathogen samples should be concentrated with the D-APDMS and DMP in a tube, and then transferred to a plate repeatedly using the centrifuge. Therefore, in the concentration step, the organic solvent is not used, and therefore cannot pass through the PTFE membrane plate. As a result, we used the PVDF membrane plate to proceed with concentration and extraction simultaneously. A total of six *E. coli* 1 mL (in PBS, 10^5^ CFU) samples were used for NAs enrichment and extraction experiments simultaneously, and qPCR was used to evaluate (Figure 5A). There was little difference in the C_T_ values of the six enriched and extracted samples, the accuracy of diagnosis has been guaranteed. To check the enrichment and extraction efficiency of the AD-DMP filter plate system, seven serial dilution *E. coli* (in PBS, 10^7^–10^1^ CFU) samples added with D-APDMS and DMP are simultaneously enriched, and the extraction experiment was conducted. When using the AD-DMP filter plate system, it was confirmed that the detection limit was 10^1^ CFU. In the same type of serially diluted samples, changes in uniform C_T_ values from low to high concentrations were confirmed (Figure 5B). To evaluate the efficiency of commercialized kits (Figure 5C) and our AD-DMP filter plate system, we confirmed the gel electrodynamics of PCR products using enriched and extracted DNA in each method. When we checked the detection limits, we could see that our system was better (Figure 5D). Our AD–DMP filter plate system can efficiently extract and concentrate samples, and on average the improved C_T_ values can also be confirmed by using qPCR and PCR (Table 2).

## 4. Conclusions

We developed a combination system, comprising an amine-functional diatomaceous earth (D-APDMS) and dimethyl pimelimidite dihydrochloride (DMP, powder) 96-well filter/membrane plate, which is simple and can simultaneously concentrate and extract several NAs from pathogen samples. By using the 96-well filter/membrane plate (0.45 µm pore size hydrophilic PVDF membrane and 0.45 µm pore size hydrophobic PTFE), we successfully enriched pathogens and then extracted NAs from multiple samples simultaneously. Compared to other systems, our AD–DMP filter/membrane plate-based method affords several advantages, with higher sensitivity due to the enrichment of pathogens from the samples than commercialized kits, including lower cost, ease of operation and manufacture, no purchase of additional laboratory equipment, and generality of use (Table 3). In addition, we demonstrated the ability of the method to simultaneously process multiple samples with various simulated matrices. Except for the enrichment step in the tube, it is possible to add reagents more quickly and easily by using a multi-pipette on the filter/membrane plate, allowing for improved time reduction. We expect it to be part of the design of a comprehensive and self-examination system. Thus, the proposed convenient multiple NAs extraction system is likely to be used for clinical applications, owing to its high performance and applicability.

## Figures and Tables

**Figure 1 micromachines-11-00823-f001:**
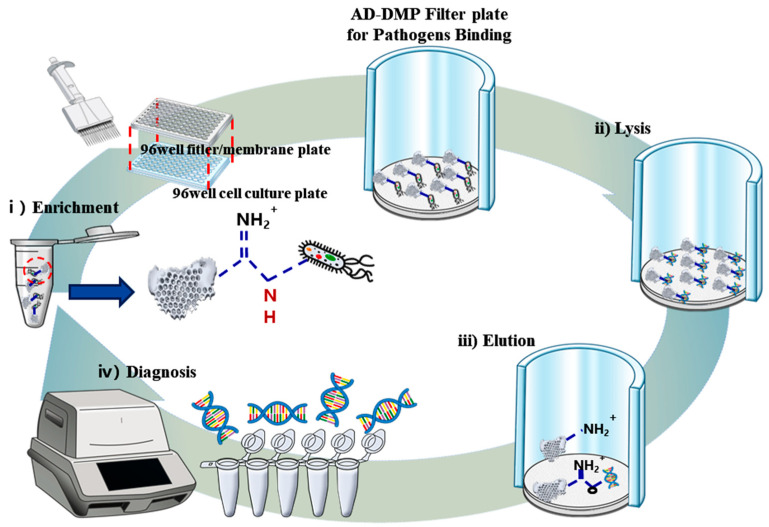
Schematic diagram of nucleic acids (NAs) enrichment and extraction via a filter/membrane plate system. Washed diatomaceous earth (DE) was modified with 3-aminopropyl-methyl-diethoxysilane (APDMS). With the help of the cross-linker dimethyl pimelimidate dihydrochloride (DMP), NAs of lysed pathogen samples can be interconnected with D-APDMS to form a solid covalent bond. The formed complex could collect NAs by using elution buffer (pH > 10).

**Figure 2 micromachines-11-00823-f002:**
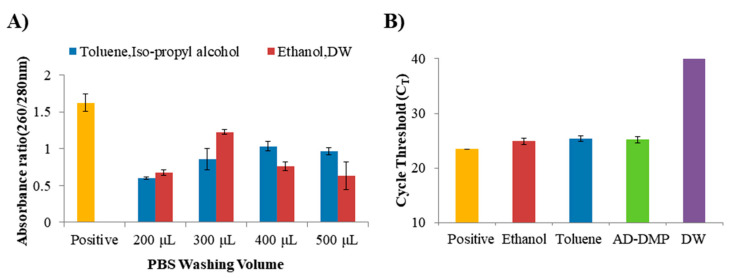
Optimization to compare the efficiency of amine-functional diatomaceous earth (D-AMDPS). (**A**) Results of progressively increasing the PBS washing volume for better nanodrop purity of D-APDMS in two ways (toluene, iso-propyl alcohol or ethanol, distilled water: DW). (**B**) D-APDMS, which was surface-treated in two ways (toluene, iso-propyl alcohol or ethanol, DW), was nucleic acids (NAs) isolated using a conventional research method, and compared with cycle threshold (C_T_) values using qPCR, along with samples separated by NAs using the newly applied amount of AD-DMP on the filter plate system. Error bars indicate standard deviation from the mean, based on at least three independent experiments.

**Figure 3 micromachines-11-00823-f003:**
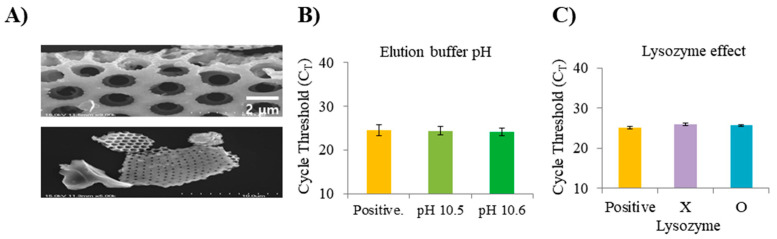
Evaluation to compare the efficiency of amine-functional diatomaceous earth (D-AMDPS). (**A**) SEM images of the surface of the D-APDMS. (**B**) C_T_ values according to various elution buffer pH. (**C**) C_T_ values according to the presence or absence of lysozyme. The C_T_ values of 40 are samples that did not detect lysozyme. Error bars indicate standard deviation from the mean, based on at least three independent experiments.

**Figure 4 micromachines-11-00823-f004:**
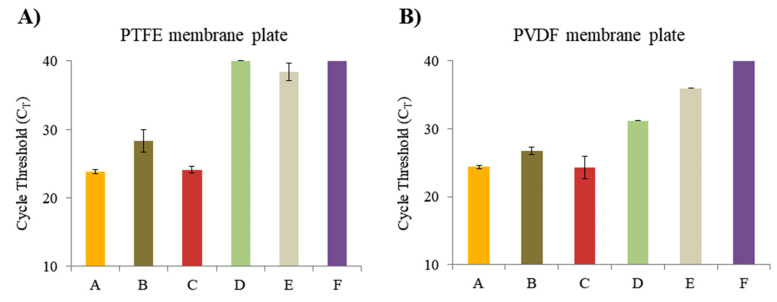
The 96-well filter/membrane plate evaluation of different properties. (**A**) Cycle threshold (C_T_) values extracted in the 96-well filter/membrane plate (0.45 µm pore size, hydrophobic polytetrafluoroethylene (PTFE)) from each different mixture in the same manner. (**B**) The C_T_ values extracted in the 96-well filter/membrane plate (0.45 µm pore size, hydrophilic PVDF membrane) from each different mixture in the same manner. A; QIAamp DNA Mini Kit. B; AD-DMP filter plate system without D-APDMS, DMP. C; AD-DMP filter plate system. D; Lysis buffer of AD-DMP filter plate system. E; AD-DMP filter plate system without *E. coli*. F; DW. It was done in the same volume using PBS. The C_T_ values of 40 are samples that did not detect. Error bars indicate standard deviation from the mean, based on at least three independent experiments.

**Figure 5 micromachines-11-00823-f005:**
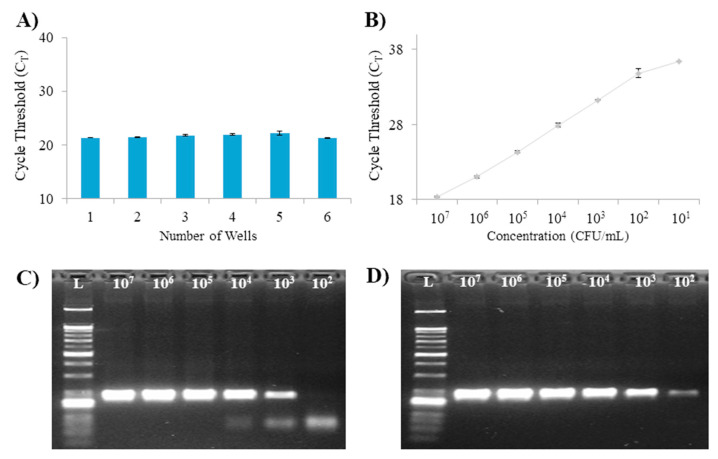
Evaluation of enrichment and extraction of several samples simultaneously in the 96-well filter/membrane plate (0.45 µm pore size, Hydrophilic PVDF membrane). (**A**) The cycle threshold (C_T_) values of the DNA enriched and extracted using the AD–DMP filter plate (0.45 µm pore size, Hydrophilic PVDF membrane) system for a 1 mL sample of the same concentration. (**B**) Evaluation of the efficiency of the AD–DMP filter plate (0.45 µm pore size, hydrophilic PVDF membrane) system for samples diluted (*E. coli* in PBS, 10^7^–10^1^ CFU/mL). Error bars indicate standard deviation from the mean, based on at least three independent experiments. (**C**) Gel electrophoresis of PCR products using DNA extracted by repeated enrichment of 1 mL samples using commercial QIAamp DNA Mini Kit (spin column, DNA Mini Kit: Cat No. 51304, Qiagen, Hilden Germany) and Centrifuge. (**D**) Gel electrophoresis of PCR products using DNA extracted by repeated enrichment of 1 mL samples using the AD–DMP filter plate (0.45 µm pore size, Hydrophilic PVDF membrane) system.

**Table 1 micromachines-11-00823-t001:** Primer sets for qPCR and PCR.

Samples	Targets	Seuqence (5′→3′)
*B. ovis*	*O 223*	F: TGG CTC GGT TGC CAA TAT CAA
		R: CGC GCT TGC CTT TCA GGT CTG
*E. coli*	*rodA-105*	F: GCA AAC CAC CTT TGG TCG
		R: CTG TGG GTG TGG ATT GAC AT

**Table 2 micromachines-11-00823-t002:** Comparison of extraction efficiencies of AD-DMP filter and column based kit.

Concentration of *E. coli*	AD-DMP Filter (C_T_ Value)	Column Based Kit (C_T_ Value)
10^7^ CFU/mL	18.31 ± 0.12	18.95 ± 0.02
10^6^ CFU/mL	21.04 ± 0.17	22.24 ± 0.10
10^5^ CFU/mL	24.35 ± 0.20	26.08 ± 0.07
10^4^ CFU/mL	27.88 ± 0.26	28.89 ± 0.02
10^3^ CFU/mL	31.28 ± 0.02	32.61 ± 0.47
10^2^ CFU/mL	34.82 ± 0.59	33.81 ± 0.75
10^1^ CFU/mL	36.45 ± 0.02	-
1 CFU/mL	-	-

**Table 3 micromachines-11-00823-t003:** Advantages and disadvantages of different NAs isolation methods.

Contents	Column-Based Kit [31]	Reported Single-Tube System [24]	Reported ADE-Filter System [12]	Multi-Filter Plate System
Number of sample available at once	One sample	One sample	One sample	6 samples
Total time	<60 min For one sample	<60 min For one sample	<60 min For one sample	<60 min For multi samples
Pathogen enrichment	No	Yes	Yes	Yes
Level of difficulty	Easy	Medium	Medium	Easy
Cost	High	Low	Low	Low
Specificity	High	High	High	High
Sensitivity	10^3^ CFU/mL	10^0^ CFU/mL	10^0^ CFU/mL	10^2^ CFU/mL
Possibility of automation	Hard and meaningless	Hard and meaningless	Possible but meaningless	Possible

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
