# Peer review of "Multi-Sample Preparation Assay for Isolation of Nucleic Acids Using Bio-Silica with Syringe Filters"

_micromachines, 2020, doi:10.3390/mi11090823_

Round 1

Reviewer 1 Report

Noh et al. present a pathogen enrichment and nucleic acid extraction strategy using functionalized diatomaceous earth, homobifunctional imidoesters, and 96-well filter/membrane plate. The idea sounds interesting and it indeed helps to address some drawbacks among regular spin-column based preparation strategies, such as multiple centrifuge steps and low through-put. However, some conclusions declared in this study are not so evidently supported. Also, on demonstrating the extraction performance of the approach, some details should be provided to make the conclusions more convincing. Therefore, a revision is necessary for finalizing the decision of acceptation for publication.

  1. In the abstract, the authors conclude that their strategy allows “faster and simpler experimentation and higher diagnosis accuracy” due to the use of multi-pipettes. I’m not sure if it’s really fast and simple, but I’m sceptical for “higher diagnosis accuracy”. In my experiences, diagnosis accuracy most depends on the downstream diagnostic methods. It doesn’t make sense to attribute it to the sample preparation. Additionally, if you think it's helpful for point-of-care diagnostics, please give some perspectives in the parts of discussions or conclusions. At least, combining PCR and real-time PCR is certainly not preferable as a point-of-care diagnostic.
  2. In the method section, when involving the extraction procedures, some details should be provided, for example, the time taken for some steps. For clarity, please do not use some undefined words, such as “a short period of time” in section 2.6.
  3. To evaluate the performance of extraction, Ct values were compared as only one parameter. However, the yield of extracted nucleic acids is also important to figure out the extraction efficiency. Please provide the total DNA yields and extraction efficiencies of your strategy and the contrast methods for Brucella ovis and Escherichia coli DNA extractions in a table.
  4. In Figure 2D, two pH values of 10.5 and 10.6 were investigated. It is confusing that why only these values are specifically examined. How about 10.2, 10.8, 11 or others? Please give some background. In addition, it shows “a slightly lower Ct value”. Why or what it means?
  5. In the text, it literally indicated Figure 3C, but it was not presented in Figure 3. Please supplement it.
  6. In Figure 4B, the error bars are hard to be noticed. Please adjust the Y range and change the bars’ colour to make them clear. Furthermore, please define error bars in the legend of each figure.
  7. How to define the limit of detection (LOD)? In Figure 3, it says “the CT values of 40 are samples that did not detect”. Based on it, in Figure 4B, it’s apparent that the Ct for detecting 10 CFU is below 40 but no Ct is indicated for testing the CFU below 10. Thus, the so-called LOD of 10 CFU will be questionable if the Ct for testing one of CFUs below 10 (e.g., 5, 2 or 1) is below 40. Similar issue for Figure 4D.   

Author Response

We thank the reviewers for their thoughtful review of our manuscript. We have taken their comments into careful consideration in preparing our revised manuscript, significantly improving its quality. Below, we present our responses to the individual comments.

Reviewer #1:

Noh et al. present a pathogen enrichment and nucleic acid extraction strategy using functionalized diatomaceous earth, homobifunctional imidoesters, and 96-well filter/membrane plate. The idea sounds interesting and it indeed helps to address some drawbacks among regular spin-column based preparation strategies, such as multiple centrifuge steps and low through-put. However, some conclusions declared in this study are not so evidently supported. Also, on demonstrating the extraction performance of the approach, some details should be provided to make the conclusions more convincing. Therefore, a revision is necessary for finalizing the decision of acceptation for publication.

In the abstract, the authors conclude that their strategy allows “faster and simpler experimentation and higher diagnosis accuracy” due to the use of multi-pipettes. I’m not sure if it’s really fast and simple, but I’m sceptical for “higher diagnosis accuracy”. In my experiences, diagnosis accuracy most depends on the downstream diagnostic methods. It doesn’t make sense to attribute it to the sample preparation.

  • We thank the reviewer for the comment.According to the reviewer’s comment, we have removed “higher diagnosis accuracy” in the revised manuscript.

Additionally, if you think it's helpful for point-of-care diagnostics, please give some perspectives in the parts of discussions or conclusions. At least, combining PCR and real-time PCR is certainly not preferable as a point-of-care diagnostic.

  • We thank the reviewer for the comment.We agreed the reviewer’s comment, so we have corrected “point-of-care diagnostic” to “clinical diagnostics” in the revised manuscript.

In the method section, when involving the extraction procedures, some details should be provided, for example, the time taken for some steps. For clarity, please do not use some undefined words, such as “a short period of time” in section 2.6.

  • We thank the reviewer for the comment.According to the reviewer’s comment, we have corrected “a short period of time” to “2 min” in the revised manuscript.

To evaluate the performance of extraction, Ct values were compared as only one parameter. However, the yield of extracted nucleic acids is also important to figure out the extraction efficiency. Please provide the total DNA yields and extraction efficiencies of your strategy and the contrast methods for Brucella ovis and Escherichia coli DNA extractions in a table.

  • We thank the reviewer for the comment.According to the reviewer’s comment, we have added Table 2 for comparison of extraction efficiencies of AD-DMP filter and conventional method in the revised manuscript.

Table 2. Comparison of extraction efficiencies of AD-DMP filter and column based kit

Concentration of E.coli

AD-DMP filter

(CT Value)

Column based kit

(CT Value)

107 CFU/mL

18.31 ± 0.12

18.95 ± 0.02

106 CFU/mL

21.04 ± 0.17

22.24 ± 0.10

105 CFU/mL

24.35 ± 0.20

26.08 ± 0.07

104 CFU/mL

27.88 ± 0.26

28.89 ± 0.02

103 CFU/mL

31.28 ± 0.02

32.61 ± 0.47

102 CFU/mL

34.82 ± 0.59

33.81 ± 0.75

101 CFU/mL

36.45 ± 0.02

-

1 CFU/mL

-

-

4In Figure 2D, two pH values of 10.5 and 10.6 were investigated. It is confusing that why only these values are specifically examined. How about 10.2, 10.8, 11 or others? Please give some background. In addition, it shows “a slightly lower Ct value”. Why or what it means?

  • We thank the reviewer for the comment.In our previous report [ref.#24], we have tested the pH value for extraction efficiency between over pH10.6 and below pH10.6. Thus, we would like to check whether small pH change might affect to extraction efficiency in pH 10.5 and pH 10.6. We have added more information on Page 8 of the revised manuscript.

On page 8, “However, when checking the CTvalue using qPCR we decided to use a buffer of pH 10.6 (CT: 24.147), showing a slightly lower CTvalue than pH 10.5 (CT: 24.407) (Fig 3(B)).”

[24] Zhao, F.; Lee, E.Y.; Shin, Y. Improved Reversible Cross-Linking-Based Solid-Phase RNA Extraction for Pathogen Diagnostics. Anal Chem 2018, 90, 1725-1733, doi:10.1021/acs.analchem.7b03493.

5In the text, it literally indicated Figure 3C, but it was not presented in Figure 3. Please supplement it.

  • We thank the reviewer for the comment.We’re sorry for the typo. We have corrected it to Fig. 4 of the revised manuscript.

In Figure 4B, the error bars are hard to be noticed. Please adjust the Y range and change the bars’ colour to make them clear. Furthermore, please define error bars in the legend of each figure.

  • We thank the reviewer for the comment.We have modified it on Figure 5B and add the information about the error bars on Figure 5B figure legend of the revised manuscript.

How to define the limit of detection (LOD)? In Figure 3, it says “the CT values of 40 are samples that did not detect”. Based on it, in Figure 4B, it’s apparent that the Ct for detecting 10 CFU is below 40 but no Ct is indicated for testing the CFU below 10. Thus, the so-called LOD of 10 CFU will be questionable if the Ct for testing one of CFUs below 10 (e.g., 5, 2 or 1) is below 40. Similar issue for Figure 4D.  

  • We thank the reviewer for the comment.In this study, when we tested the primer sets of the targets (coli or Brucella Ovis) either with positive or negative samples, the negative control with distilled water is also amplified over 40 ct value by RT-PCR. Thus, 40 Ct value was used as a criterion to decide positive or negative sample. When we have tested the concentration of bacteria below 10 CFU, it could not be detected properly within 40 Ct value.

Reviewer 2 Report

Overview:

In this work, the authors develop a platform for parallelized extraction of nucleic acids from multiple bacterial samples. The authors use a 96-well cell culture plate as the base of their platform, and use amine-functionalized particles of diatomaceous earth (DE) for solid-phase capture of either bacterial cells or bacterial nucleic acids. These DE particles are then trapped by a filter membrane which is interfaced into the 96-well cell culture plate, and the captured nucleic acids (after enrichment and lysis, in the case of bacterial cells) can be subsequently eluted, spun down, and collected for PCR-based detection in individual tubes. The authors’ motivation in developing their platform is to provide a faster, less-expensive, less-laborious, and higher-throughput alternative to the washing and tube-transfer-intensive column-based serial nucleic acid extraction process. While the authors finally do show similar, if not slightly better, limits of detection using their novel methodology than traditional column-based methods, there are significant concerns in the clarity and presentation of the work that should require major revisions before reconsideration. Furthermore, the authors should address the following major and minor concerns for effectively clarifying the attributes of their platform.

Major concerns:

  • In many instances the manuscript suffers from a lack of clarity, and much of it can be attributed to the writing. The authors will benefit from having a native speaker proofread the manuscript for extensive language and grammatical errors. To provide only a few examples, in line 32, the sentence “Thus, not only demanding…” requires rework. In page 2, line 50, “introspective system” is incorrectly used. The sentence starting on line 243 “The new amine-functionalized method…” makes it unclear which is the published/established method and which is the modified one in this work. A thorough revision of the text is warranted to enhance clarity.
  • The motivation behind the development of the proposed platform is to enhance the throughput and decrease the laboriousness of traditional nucleic acid extraction. However in their method, the authors require many user-initiated steps. For example, the processes of trapping pathogen cells bound to DE onto the PVDF filter requires at least 4 pipetting and centrifugations steps before lysis (along with discarded tubes containing flow-through). Furthermore, before elution, 2 washing steps are required. Such tedious pipetting steps, while parallelizable, do not pose a significant advantage over traditional column-based methods. The authors should be more clear about their exact advantages over column-based methods.
  • The authors should offer some explanation, at least a brief one, into the chemistry/mechanisms underlying pathogen capture and nucleic capture by amine-functionalized diatomaceous earth, and the apparent enhancement caused by the addition of HI or DMP
  • Please modify axes labels for clarity in Figure 2 (e.g., “PBS washing” should read “PBS wash volume” and “260/280” should read “Absorbance Ratio (260 nm/280 nm)”)
  • The authors use the term “micro-gridder” DE without providing any previous explanation as to what this is
  • There is a reference to a Figure 3(C) in the text (line 307) but no corresponding figure
  • In the Abstract, it is claimed that more than 10 samples were processed simultaneously. In Introduction, it is claimed that at least 8 samples can be processed simultaneously; yet, the data shows only 6 samples at most. The authors should have a consistent, verifiable claim.
  • Report linearity in 4(B); report units on graphs
  • In the Introduction, the authors claim “two to four times more sensitivity,” yet do not present data to back up this numerical claim.

Minor concerns:

  • Fix wording in Table 2 – “multi samples” to “6 samples,” “worthy expected” to “expected” or “possible” or something more appropriate
  • Make sure that all instances of bacterial species are italicized
  • From Figure 2A, how do the authors conclude that ethanol-based amine functionalization is better than toluene-based amine functionalization? The data (at least for the higher PBS washes) does not indicate this. Why?
  • Line 271: report average Ct value – it does not look like pH10.6 is different than pH10.5

Author Response

We thank the reviewers for their thoughtful review of our manuscript. We have taken their comments into careful consideration in preparing our revised manuscript, significantly improving its quality. Below, we present our responses to the individual comments.

Reviewer #2:

In this work, the authors develop a platform for parallelized extraction of nucleic acids from multiple bacterial samples. The authors use a 96-well cell culture plate as the base of their platform, and use amine-functionalized particles of diatomaceous earth (DE) for solid-phase capture of either bacterial cells or bacterial nucleic acids. These DE particles are then trapped by a filter membrane which is interfaced into the 96-well cell culture plate, and the captured nucleic acids (after enrichment and lysis, in the case of bacterial cells) can be subsequently eluted, spun down, and collected for PCR-based detection in individual tubes. The authors’ motivation in developing their platform is to provide a faster, less-expensive, less-laborious, and higher-throughput alternative to the washing and tube-transfer-intensive column-based serial nucleic acid extraction process. While the authors finally do show similar, if not slightly better, limits of detection using their novel methodology than traditional column-based methods, there are significant concerns in the clarity and presentation of the work that should require major revisions before reconsideration. Furthermore, the authors should address the following major and minor concerns for effectively clarifying the attributes of their platform.

Major concerns:

  • In many instances the manuscript suffers from a lack of clarity, and much of it can be attributed to the writing. The authors will benefit from having a native speaker proofread the manuscript for extensive language and grammatical errors. To provide only a few examples, in line 32, the sentence “Thus, not only demanding…” requires rework.
  • We thank the reviewer for the comment.When we submit this manuscript at first, the manuscript have been proofread by English native speakers from Editage, where is one of English editing company. We provide the certification from Editage.

  • Nevertheless, we have modified the sentence on Page 2 of the revised manuscript.

On page 2, “Thus,rapid diagnostics techniques combined with automation and/or self-examination system areincreasingly expected to save labor as well as time and provide better protection of the medical inspectors[5].”

In page 2, line 50, “introspective system” is incorrectly used.

  • We have removed the sentence on Page 3 of the revised manuscript.

On page 2, “A comprehensive system,……..”

The sentence starting on line 243 “The new amine-functionalized method…” makes it unclear which is the published/established method and which is the modified one in this work. A thorough revision of the text is warranted to enhance clarity.

  • We have modified the sentence on Page 7 of the revised manuscript.

On page 7, “The DE amine-functionalized method in this study using toluene and iso-propyl alcohol is compared with the laboratory method using ethanol and DW. The DE amine-functionalized by two methods were used for NAs extraction in the same previously reported experimental method..”

  • The motivation behind the development of the proposed platform is to enhance the throughput and decrease the laboriousness of traditional nucleic acid extraction. However in their method, the authors require many user-initiated steps. For example, the processes of trapping pathogen cells bound to DE onto the PVDF filter requires at least 4 pipetting and centrifugations steps before lysis (along with discarded tubes containing flow-through). Furthermore, before elution, 2 washing steps are required. Such tedious pipetting steps, while parallelizable, do not pose a significant advantage over traditional column-based methods. The authors should be more clear about their exact advantages over column-based methods.
  • We thank the reviewer for the comment. The strongest advantage of the proposed assay is able to enrich pathogen and extract nucleic acid simultaneously from the large volume of samples. No existing assay can do it at once from the samples. Therefore, we have added the information on conclusions and Table 2 of the revised manuscript.

On Page 12, “By using the 96-well filter/membrane plate (0.45 µm pore size hydrophilic PVDF membrane and 0.45 µm pore size hydrophobic PTFE), we successfully enriched pathogen and then extracted NAs from multiple samples simultaneously. Compared to other systems, our AD-DMP filter/membrane plate-based method affords several advantages, with higher sensitivity due to the enrichment of pathogen from the samples than commercialized kits, including lower cost, ease of operation and manufacture, no purchase of additional laboratory equipment, and generality of use (Table 2).”

  • The authors should offer some explanation, at least a brief one, into the chemistry/mechanisms underlying pathogen capture and nucleic capture by amine-functionalized diatomaceous earth, and the apparent enhancement caused by the addition of HI or DMP.
  • We thank the reviewer for the comment. As we described in the introduction part, we have published previously regarding the chemical principle of the DE and HI binding with pathogen and nucleic acids. According to the reviewer’s comment, we have added more information on Page 2 ofthe revised manuscript.

On page 2, “The positive charge of the amine-modified DE surface can bind with the negative charge of pathogen surface by electrostatic interaction [23]. Meanwhile, an additional homobifunctional imidoester (HI) reagent was used to combine more powerfully with NAs. The amine group of HI surface can bind with the amine group of NA by covalent bonding [23,24].”

  • Please modify axes labels for clarity in Figure 2 (e.g., “PBS washing” should read “PBS wash volume” and “260/280” should read “Absorbance Ratio (260 nm/280 nm)”)
  • We thank the reviewer for the comment.According to the reviewer’s comment, we have modified those on Figure 2A of the revised manuscript.

  • The authors use the term “micro-gridder” DE without providing any previous explanation as to what this is
  • We thank the reviewer for the comment.We have modified it in the revised manuscript.

“micro-gridder DE” is changed to “DE having micro-grid structure” on page 2 and 8 of the revised manuscript.

  • There is a reference to a Figure 3(C) in the text (line 307) but no corresponding figure
  • We thank the reviewer for the comment.It is a typo. We have corrected it to Fig. 4 of the revised manuscript.

  • In the Abstract, it is claimed that more than 10 samples were processed simultaneously. In Introduction, it is claimed that at least 8 samples can be processed simultaneously; yet, the data shows only 6 samples at most. The authors should have a consistent, verifiable claim.
  • We thank the reviewer for the comment.According to the reviewer’s comment, we have changed it to 6 samples on Page 1 of the revised manuscript.

  • Report linearity in 4(B); report units on graphs
  • We thank the reviewer for the comment.We have modified it on Figure 5B of the revised manuscript.

  • In the Introduction, the authors claim “two to four times more sensitivity,” yet do not present data to back up this numerical claim.
  • We thank the reviewer for the comment.Based on the results of Figure 5 and Table 2, the proposed assay showed 10 times more sensitivity (101CFU/mL) as a result of RT-PCR compared to commercialized NA kits (102CFU/mL). We have corrected it on the introduction of the revised manuscript.

Minor concerns:

  • Fix wording in Table 2 – “multi samples” to “6 samples,” “worthy expected” to “expected” or “possible” or something more appropriate
  • We thank the reviewer for the comment.We have modified those on Table 3 of the revised manuscript.

  • Make sure that all instances of bacterial species are italicized
  • We thank the reviewer for the comment.We have modified it in the revised manuscript.

  • From Figure 2A, how do the authors conclude that ethanol-based amine functionalization is better than toluene-based amine functionalization? The data (at least for the higher PBS washes) does not indicate this. Why?
  • We thank the reviewer for the comment.According to the results of the Nanodrop, the average purity was the best when the PBS washing was 300 μL in the ethanol-based amine functionalization (Fig. 2(A)). Although the average purity of the toluene-based amine functionalization at the higher PBS washing was better than that of the ethanol-based amine functionalization, the higher PBS washing required more operation time. This study is purposed for development of rapid multiple sample preparation assay. Therefore, the ethanol-based amine functionalization method was used for this proposed assay.

  • Line 271: report average Ct value – it does not look like pH10.6 is different than pH10.5
  • We thank the reviewer for the comment.We have added the information on Page 8 ofthe revised manuscript.
  • On page 8, “However, when checking the CTvalue using qPCR we decided to use a buffer of pH 10.6 (CT: 24.147), showing a slightly lower CTvalue than pH 10.5 (CT: 24.407) (Fig 3(B)).”

Round 2

Reviewer 1 Report

The revised version is satisfactory since some details have been presented.  

Reviewer 2 Report

The authors have satisfactorily addressed all the provided comments.

However, a few suggested improvements remain:

  • In the abstract, line 21, modify "more than 6 samples" to "as many as 6 samples" (please correct the actual value based on your results. It is still unclear if the number is 7 (line 334) or 6 (line 330) or 8 (line 85))
  • change the word "multi-pipettes" to "multichannel pipettes"
  • line 76, modify "substance area" to "surface area"
  • line 256, "nanodrop" -> "DNA absorbance"
  • line 289, "base" -> "based"
  • Table 3: "Single" -> "1 sample"; "6 samples" -> value based on first comment; 
  • line 369 "biological matrices" -> "buffer" or "simulated matrices"